# Prognostic Effect of Preoperative Psoas Muscle Hounsfield Unit at Radical Cystectomy for Bladder Cancer

**DOI:** 10.3390/cancers13225629

**Published:** 2021-11-10

**Authors:** Yusuke Sugino, Takeshi Sasaki, Manabu Kato, Satoru Masui, Kouhei Nishikawa, Takashi Okamoto, Shinya Kajiwara, Takuji Shibahara, Takehisa Onishi, Shiori Tanaka, Hideki Kanda, Hiroshi Matsuura, Takahiro Inoue

**Affiliations:** 1Department of Nephro-Urologic Surgery and Andrology, Mie University Graduate School of Medicine, 2-174 Edobashi, Tsu 514-8507, Mie, Japan; y-sugino@med.mie-u.ac.jp (Y.S.); t-sasaki@med.mie-u.ac.jp (T.S.); katouro@med.mie-u.ac.jp (M.K.); s-masui@med.mie-u.ac.jp (S.M.); kouheini@med.mie-u.ac.jp (K.N.); 2Department of Urology, Ise Red Cross Hospital, 1-471-2 Funae, Ise 516-8512, Mie, Japan; t.okamoto555.medic@gmail.com (T.O.); shinya6226@gmail.com (S.K.); shibaharauro@chive.ocn.ne.jp (T.S.); takehisa@ise.jrc.or.jp (T.O.); 3Mie Prefectural General Medical Center, Department of Urology, 5450-132 Hinaga, Yokkaichi 510-8561, Mie, Japan; t-shiori@med.mie-u.ac.jp (S.T.); hideki-kanda@mie-gmc.jp (H.K.); hiroshi-matsuura@mie-gmc.jp (H.M.)

**Keywords:** bladder cancer, urothelial carcinoma, radical cystectomy, frailty, prognostic factor, psoas muscle, Hounsfield units

## Abstract

**Simple Summary:**

Radical cystectomy (RC) is the standard treatment for patients with advanced bladder cancer. Since RC is a highly invasive procedure, it is necessary to carefully predict the prognosis before surgery and to determine the surgical indication. According to the results of the retrospective analysis of our 177 RC cases, we found the Hounsfield units of the psoas muscle at the third lumbar vertebral level to be a prognostic factor. Univariate and multivariate analyses revealed that age, sex, clinical T stage, and psoas muscle Hounsfield units were significant preoperative factors for overall survival. Furthermore, risk classification using these four factors was useful for predicting the prognosis of patients with RC.

**Abstract:**

Radical cystectomy (RC) is the standard treatment for patients with advanced bladder cancer. Since RC is a highly invasive procedure, the surgical indications in an aging society must be carefully judged. In recent years, the concept of “frailty” has been attracting attention as a term used to describe fragility due to aging. We focused on the psoas muscle Hounsfield unit (PMHU) and analyzed its appropriateness as a prognostic factor together with other clinical factors in patients after RC. We retrospectively analyzed the preoperative prognostic factors in 177 patients with bladder cancer who underwent RC between 2008 and 2020. Preoperative non-contrast computed tomography axial image at the third lumbar vertebral level was used to measure the mean Hounsfield unit (HU) and cross-sectional area (mm^2^) of the psoas muscle. Univariate analysis showed significant differences in age, sex, clinical T stage, and PMHU. In multivariate analysis using the Cox proportional hazards model, age (hazard ratio (HR) = 1.734), sex (HR = 2.116), cT stage (HR = 1.665), and PMHU (HR = 1.758) were significant predictors for overall survival. Furthermore, using these four predictors, it was possible to stratify the prognosis of patients after RC. Finally, PMHU was useful as a simple and significant preoperative factor that correlated with prognosis after RC.

## 1. Introduction

Radical cystectomy (RC) is the gold standard treatment for patients with muscle invasive bladder cancer, patients with selected T1 high-grade non-muscle invasive bladder cancer, and patients with carcinoma in situ resistant to Bacillus Calmette–Guérin treatment [1]. RC remains one of the most invasive urological procedures, and its surgical indication needs to be carefully assessed in an aging society. According to the “Annual Report on the Aging Society 2020” from the Cabinet Office of Japan, the total population of Japan is 126.17 million as of 2019, of which 35.89 million are aged 65 years or older. Japan is facing an aging society ahead of the rest of the world. A systematic review has reported that perioperative mortality within 90 days after RC significantly increases in the elderly, and overall survival (OS) and cancer-specific survival (CSS) also decrease with age [2]. Aging is clearly a risk factor for RC. In recent years, the concept of “frailty” has been attracting attention as a term that expresses fragility due to aging [3]. If the prognosis can be predicted more accurately by assessing not only the clinical stage and age of the patient but also malnutrition and muscle weakness associated with decreased physical activity, it can help in deciding whether to perform surgery. Although few established definitions with regard to the elderly or frailty have been reported, there have been some attempts to define them with various assessments [3,4,5]. Several reports linking frailty and sarcopenia to predict prognosis in patients with bladder cancer have been reported [3,6,7,8,9,10]. As an indicator that may objectively represent frailty, we focused on the psoas muscle Hounsfield unit (PMHU), which is defined as the mean computed tomographic attenuation value of the psoas muscle. The main aim of this study is to assess the utility of PMHU as a preoperative prognostic marker in patients receiving RC for bladder cancer.

## 2. Materials and Methods

### 2.1. Study Design and Patients

We retrospectively reviewed the records of consecutive patients who underwent open radical cystectomy (ORC), laparoscopic radical cystectomy (LRC), or robot-assisted laparoscopic radical cystectomy (RARC) for bladder cancer at Mie University Hospital, Ise Red Cross Hospital, and Mie Prefectural General Medical Center. A total of 177 patients (113 patients at Mie University Hospital, 42 patients at Ise Red Cross Hospital, and 23 patients at Mie Prefectural General Medical Center) were enrolled.

### 2.2. Image Analyses

Abdominal non-contrast computed tomography (NCCT) images (1–5 mm-thick slices) taken within 3 months before RC were used to measure the imaging factors related to the psoas muscle. Four urologists (Y.S., S.K., T.O., and S.T.) freehand outlined each psoas muscle at the third lumbar vertebral level in the axial NCCT image (Figure 1).

The right and left total areas were used for the psoas muscle area (PMA) (mm^2^). The mean value of the mean computed tomographic attenuation value of the right and left psoas muscle was used for the PMHU (HU). The psoas mass index (PMI) (cm^2^/m^2^) was calculated by normalizing the cross-sectional area by height [11].

### 2.3. Statistical Analyses

All statistical analyses were performed using EZR version 1.33 [12]. Student’s *t*-test or Mann–Whitney U test was performed for comparisons between groups of continuous variables. Categorical variables were analyzed using the chi-squared test or Fisher’s exact test. The survival curve was estimated using the Kaplan–Meier method and analyzed using the log-rank test. Cox proportional hazards analysis was used to calculate the hazard ratio (HR) and 95% confidence interval (CI) in univariate and multivariate analyses. In all tests, *p* < 0.05 was considered statistically significant.

## 3. Results

Of the 177 patients, 26 (14.7%) were women and 151 (85.3%) were men. The median age was 70 (quartile: 66–76) years, and the median follow-up period was 1002 (quartile: 358–1989) days. The 5-year OS, CSS, and recurrence-free survival (RFS) rates were 59.7%, 71.3%, and 48.7%, respectively. The median OS, CSS, and RFS were 8.86 years (95% CI, 5.00 years to not reached), not reached (95% CI, not reached to not reached), and 4.70 years (95% CI, 2.66 to 9.24 years), respectively. Platinum-based neoadjuvant and adjuvant chemotherapy were performed in 75 (42.4%) and 22 (12.4%) patients, respectively. The surgical procedures were ORC, LRC, and RARC in 119 (67.2%), 38 (21.5%), and 20 (11.3%) patients, respectively. There were 114 (64.4%) and 29 (16.4%) perioperative complications of Clavien–Dindo grade ≥2 and ≥3, respectively. The median length of hospital stay after RC was 26 (quartile: 22–41) days. The histopathological diagnosis after RC was urothelial carcinoma in 138 (77.8%) patients, and some histological variants were found in 39 (22.0%) patients. There were 71 (40.1%) patients with pT ≥3, 33 (18.6%) patients with pN positivity, and 89 (50.3%) patients with LVI positivity, all of which were significant prognostic factors for OS (*p* < 0.01, log-rank test). Body mass index, PMI, PMA, and PMHU were significantly different between men and women (Table 1).

For variables related to the psoas muscle, it was considered inappropriate to apply the same cutoff value for men and women; thus, the lower limit of the interquartile range for each sex (25 percentile) was used as the cutoff value.

The results of Cox proportional hazards analysis for OS were shown in Table 2.

Univariate analysis showed significant differences in age, sex, cT stage, and PMHU (*p* < 0.05). The median OS stratified by PMHU alone in the not-low and low PMHU groups were 9.24 years (95% CI, 6.40 years to not reached) and 2.78 years (2.06 years to not reached), respectively, and there was a significant difference among them (*p* = 0.014). Multivariate analysis using the four factors that were significantly different in the univariate analysis showed significant differences in all factors (*p* < 0.05).

We focused on these four factors to develop a risk classification for predicting OS in patients with bladder cancer after RC (Table 3). The Kaplan–Meier curve for OS according to the number of risks was shown in Figure 2.

Based on this result, we defined a group with one or fewer risk factors as a low-risk group, a group with two risk factors as an intermediate-risk group, and a group with three or more risk factors as a high-risk group (Table 3). The Kaplan–Meier curve and HR for each risk category were shown in Figure 3 and Table 4, respectively.

The median OS by our risk category in the low-risk, intermediate-risk, and high-risk groups (Table 3) were not reached (95% CI, 8.86 years to not reached), 6.40 years (95% CI, 2.67 years to not reached), and 2.06 years (95% CI, 0.94 to 2.78 years), respectively (*p* < 0.01). There were no significant differences among the risk groups in terms of postoperative hospital stay and the incidence of complications.

## 4. Discussion

In the present study, we investigated the significance of preoperative PMHU on the prognosis after RC in patients with bladder cancer. We showed that PMHU is a new prognostic marker along with sex, age, and cT stage, and that these predictors could be used to preoperatively stratify the prognosis of patients with bladder cancer after RC.

In recent years, medical technology innovations have led to minimally invasive surgery, but RC remains highly invasive because of the long operation time and high incidence of perioperative complication rates [2]. Elderly people had a higher 90-day mortality rate and more early complications after RC than younger people [2], and it is often discussed whether RC should be performed, especially for elderly patients.

In order to judge the indication for RC, it is required not only to evaluate the surgical tolerance such as cardiac function and respiratory function, but also to predict the postoperative prognosis to some extent. LVI and pN positivity and pT3 or higher grade have already been reported as postoperative factors related to prognosis [13], and similar results were obtained with our cases. However, pathological factors are the information that can only be known after RC. There are few reports on prognostic markers that are useful in deciding whether or not to perform RC itself.

In an aging society, the concept of frailty is drawing attention. Especially in the elderly, comprehensive assessment of patients’ frailty as well as their age may be useful for treating diseases and improving their quality of life [14,15]. Diagnosis of frailty requires the measurement of grip strength and walking speed [4,5], but there are few facilities that can be incorporated into the daily practice of treating patients with urological malignancies. In particular, it is not desirable to impose a heavy preoperative evaluation on patients with their malignancies.

Regarding the assessment of frailty, NCCT images are easy to acquire and non-invasive. PMA [16], psoas muscle volume (PMV) [9,17,18], PMI [9,11], mean PMHU [19], skeletal muscle index (SMI) [6,8,20], intramuscular adipose tissue content (IMAC) [21], and so on, have been reported as representative imaging factors for frailty. According to a systematic review by Cao et al., SMI, PMI, muscle attenuation, and IMAC were useful for assessing the risk of postoperative complications as NCCT-assessed sarcopenia indices [22]. Analysis of our data did not reveal significant results for PMA and PMI, but PMHU correlated with the prognosis of patients with bladder cancer after RC.

PMHU can be measured very simply and easily. No special training or software is required. NCCT images are always acquired in patients before RC, without the burden of adding new special tests to the patients. Low PMHU reflects skeletal muscle fat infiltration and may indirectly be used to assess frailty [23]. Increase in fat infiltration within skeletal muscle might precede loss of skeletal muscle volume during the progression of cancer cachexia [24].

PMHU is a factor that reflects muscle quality, while PMA and PMI are factors that reflect muscle mass [20,22]. In our data, this may be one of the reasons why only PMHU, not PMA and PMI, showed a significant correlation with OS. Assessing psoas muscle mass with an NCCT image at the L3 level is very simple but may not necessarily reflect systemic skeletal muscle mass [25]. PMV and SMI may reflect systemic muscle mass better than PMA and PMI, but at the expense of ease of measurement [18]. We consider PMHU to be a more practical predictor because of its ease of measurement and its accuracy for prognosis.

Frailty was also evaluated as a risk factor for perioperative complications in patients with bladder cancer [9,26]. Prediction of patients’ prognosis preoperatively may avoid surgical invasive procedures which would cause more harm than benefit in patients. According to our results, the median OS of high-risk patients who have three or more preoperative poor prognosis factors defined by age, sex, cT stage, and PMHU was only 2.06 years. Although our data would not give definitive prediction of perioperative complication, high-risk patients with poor prognosis in our risk classification might not recommend RC. Bladder sparing therapy combined with transurethral resection of the bladder tumor, chemotherapy, and radiation therapy are controversial, but they may be a good treatment option for patients with an apparently poor prognosis [9,27].

By preoperatively diagnosing frailty, it may be possible to improve the prognosis of cancer patients if interventions such as exercise therapy and nutritional guidance can be performed earlier [17,28]. Exercise therapy and essential amino acid supplement drinks have been shown to be useful in recovery from serious illnesses [29] and vitamin D supplementation was useful in improving sarcopenia in the elderly [30]. It goes without saying that treatment of the bladder cancer itself is important for improving the OS. However, in an aging society, we must understand the physical function and nutritional status of each patient before intervening with diverse treatments. To that end, the role of the rehabilitation team, which includes registered dietitians and physiotherapists, is also important, and more than ever it is necessary to deepen cooperation within the team.

There are several limitations of our study. The present study is a small retrospective study. Our results need to be prospectively validated in a larger cohort. In addition, the subjects of this study were limited to the Japanese population, and there is room for consideration of differences between races, especially regarding the cutoff value. Furthermore, the present study did not directly investigate the relationship between PMHU and frailty. In the future, the direct relationship between the already reported diagnostic factors of frailty and PMHU must be investigated [4,5]. In addition we believe that it should be examined in more detail whether perioperative nutrition therapy and physiotherapy for patients with low PMHU can improve the bladder cancer patients’ survival and quality of life.

## 5. Conclusions

PMHU was a preoperative predictor of prognosis in patients with bladder cancer who were about to undergo RC. The prognosis of patients could be stratified before RC using age, sex, cT stage, and PMHU. Not only can PMHU be measured without burdening patients and clinicians, but also this risk classification helps determine whether to perform RC in patients with bladder cancer before surgery.

## Figures and Tables

**Figure 1 cancers-13-05629-f001:**
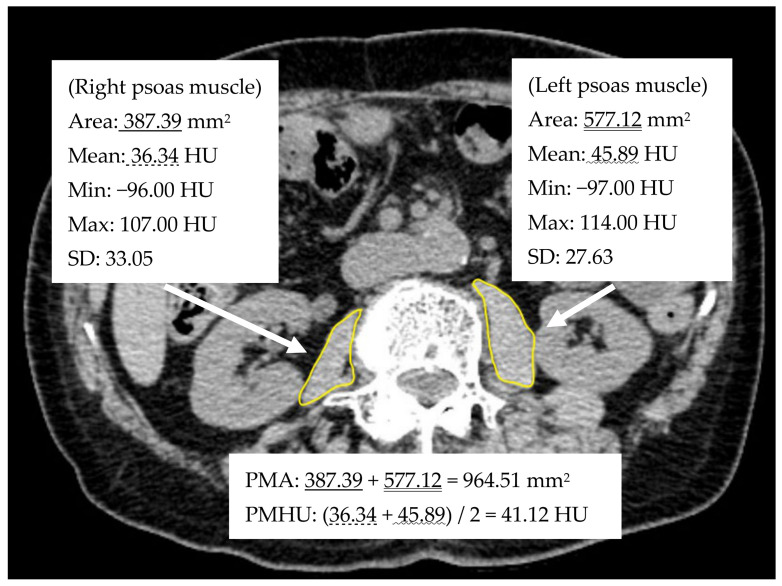
Measurement of the psoas muscle area (PMA) and the mean psoas muscle Hounsfield unit (PMHU) on the preoperative axial non-contrast computed tomography image at the third lumbar vertebral level.

**Figure 2 cancers-13-05629-f002:**
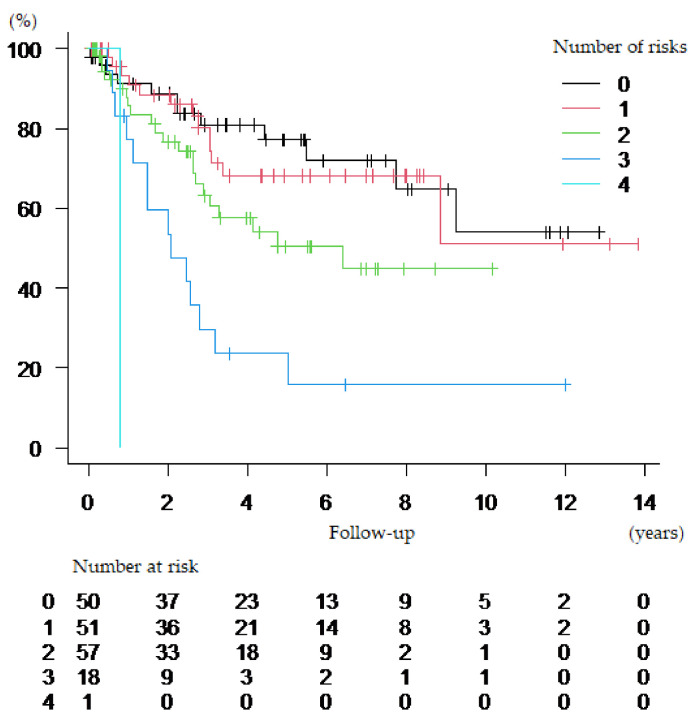
Kaplan–Meier curves for overall survival according to the number of risks.

**Figure 3 cancers-13-05629-f003:**
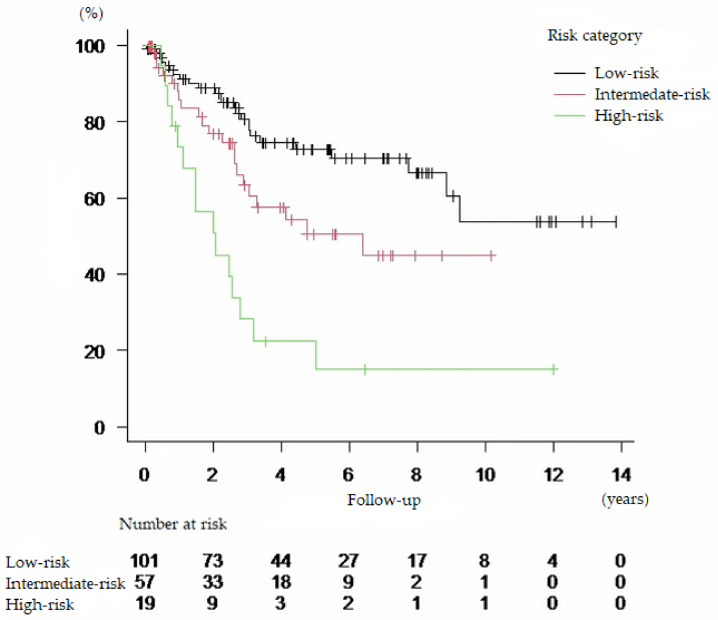
Kaplan–Meier curves for overall survival according to risk category.

**Table 1 cancers-13-05629-t001:** Patient characteristics.

Variables		Total Cases (*n* = 177)	Men (*n* = 151)	Women (*n* = 26)	*p*-Value
Median age at RC (IQR)		70 (66–76)	70 (66–76)	72.5 (65–76)	0.687
Median BMI (IQR)		22.92 (20.94–24.95)	23.11 (21.20–25.07)	21.22 (19.03–22.68)	<0.01
ASA-PS	1	21 (12%)	17 (11%)	4 (15%)	0.755
	2	120 (68%)	103 (68%)	17 (65%)	
	3	36 (20%)	31 (21%)	5 (19%)	
Clinical T stage, *n* (%)	NMIBC	51 (29%)	45 (30%)	6 (23%)	0.163
	2	70 (40%)	61 (40%)	9 (35%)	
	3	27 (15%)	19 (13%)	8 (31%)	
	4	29 (16%)	26 (17%)	3 (12%)	
Clinical N stage, *n* (%)	0	154 (87%)	132 (87%)	22 (85%)	0.752
	≥1	23 (13%)	19 (13%)	4 (15%)	
Neoadjuvant chemotherapy, *n* (%)	No	102 (58%)	90 (60%)	12 (46%)	0.207
	Yes	75 (42%)	61 (40%)	14 (54%)	
Median PMA (mm^2^) (IQR)		1129 (894–1455)	1242 (983–1510)	679 (521–887)	<0.01
Median PMI (cm^2^/m^2^) (IQR)		4.31 (3.41–5.41)	4.66 (3.66–5.60)	2.88 (2.34–3.89)	<0.01
Median PMHU (HU) (IQR)		43.14 (39.26–47.54)	43.40 (39.56–47.75)	40.93 (34.56–45.16)	0.043

IQR = interquartile range, BMI = body mass index, ASA-PS; American Society of Anesthesiologists physical status, NMIBC = non-muscle invasive bladder cancer, PMA = psoas muscle area, PMI = psoas mass index, PMHU = psoas muscle Hounsfield unit.

**Table 2 cancers-13-05629-t002:** Cox proportional hazards analysis of overall survival in patients undergoing radical cystectomy.

		Univariate			Multivariate		
Variables	Category	HR	95% CI	*p* Value	HR	95% CI	*p* Value
Age	<70	Reference			Reference		
	≥70	2.093	1.239–3.533	0.006	1.734	1.010–2.977	0.046
Sex	Men	Reference			Reference		
	Women	2.210	1.189–4.109	0.012	2.116	1.132–3.954	0.019
ASA-PS	1,2	Reference					
	3	1.624	0.926–2.849	0.091			
Clinical T stage	<3	Reference			Reference		
	≥3	1.782	1.705–2.956	0.025	1.665	1.001–2.769	0.049
Clinical N stage	0	Reference					
	≥1	0.659	0.283–1.530	0.33			
Neoadjuvant chemotherapy	No	Reference					
	Yes	0.877	0.512–1.502	0.633			
BMI	Not low	Reference					
	Low	0.909	0.507–1.631	0.749			
PMA	Not low	Reference					
	Low	1.332	0.692–2.564	0.391			
PMI	Not low	Reference					
	Low	1.019	0.530–1.959	0.954			
PMHU	Not low	Reference			Reference		
	Low	1.924	1.132–3.270	0.016	1.758	1.014–3.048	0.044

HR = hazard ratio, CI = confidence interval, ASA-PS = American Society of Anesthesiologists physical status, BMI = body mass index, PMA = psoas muscle area, PMI = psoas mass index, PMHU = psoas muscle Hounsfield unit.

**Table 3 cancers-13-05629-t003:** Risk factors and risk category.

Risk Factors	0	1
1. Age	<70	≥70
2. Sex	Men	Women
3. Clinical T stage	<3	≥3
4. PMHU	Men: ≥39.56 HU Women: ≥34.56 HU	Men: <39.56 HU Women: <34.56 HU
**Risk Category**		
Low-risk	If <2 risk factors present	
Intermediate-risk	If 2 risk factors present	
High-risk	If ≥3 risk factors present	

PMHU = psoas muscle Hounsfield unit.

**Table 4 cancers-13-05629-t004:** Hazard ratio for each risk category.

Risk Category	*n*	HR	95% CI	*p* Value
Low-risk	101	Reference		
Intermediate-risk	57	1.902	1.061–3.411	0.031
High-risk	19	4.597	2.408–8.775	<0.01

HR = hazard ratio, CI = confidence interval.

## Data Availability

The data are available to other researchers on written request to the corresponding author.

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
