# Peer review of "Prognostic Effect of Preoperative Psoas Muscle Hounsfield Unit at Radical Cystectomy for Bladder Cancer"

_cancers, 2021, doi:10.3390/cancers13225629_

Round 1

Reviewer 1 Report

This manuscript describes the potential of psoas muscle Hounsfield unit (PMHU) as a prognostic factor together with other clinical characteristics in patients after RC. The manuscript is well-written, but some issues need to be addressed.

  • The Introduction section should include the definition of PMHU and state a clear aim.
  • The Material and methods section should be divided into sub-sections.
  • What is the performance of PMHU alone as a prognostic factor (in terms of overall survival)?
  • On page 8 (lines 210-213), the Authors say that "According to our results, the 2-year OS rate and 5-year OS rate of high-risk patients who have three or more preoperative poor prognosis factors defined by age, sex, cT stage, and PMHU are 50% and 20%, respectively.". However, only a 5-year OS rate is presented in the Results Section (page 7, lines 157-159), and the value is not the same (20% vs. 22.6%).

Author Response

Response to reviewers:

We should thank the reviewers for these precious comments concerning my manuscript entitled Prognostic effect of preoperative psoas muscle Hounsfield unit at radical cystectomy for bladder cancer. These comments are all valuable and extremely helpful for revising and improving our paper. We have checked comments carefully and have made corrections which we hope meet with approval. The responses to the reviewer’s comments are as follows:

Reviewer 1

  • The Introduction section should include the definition of PMHU and state a clear aim.

Our response: Thank you for pointing this out. We have revised the Introduction section to include a definition of PMHU and the aims of this study. 

  • The Material and methods section should be divided into sub-sections.

Our response: Thank you for pointing this out. We have divided the Material and methods into sub-sections.

  • What is the performance of PMHU alone as a prognostic factor (in terms of overall survival)?

Our response: Thank you very much for these valuable comments. We have revised the Result section to include the performance of PMHU alone and each risk category for median OS.

  • On page 8 (lines 210-213), the Authors say that "According to our results, the 2-year OS rate and 5-year OS rate of high-risk patients who have three or more preoperative poor prognosis factors defined by age, sex, cT stage, and PMHU are 50% and 20%, respectively.". However, only a 5-year OS rate is presented in the Results Section (page 7, lines 157-159), and the value is not the same (20% vs. 22.6%).

Our response: Thank you for pointing this out. The 5-year OS rate of 20% was incorrect. For brevity, we have described the median OS rather than the 5-year OS rate for the Result and the Discussion.

Reviewer 2 Report

The authors in their publication analyse some risk factors in patients undergoing radical cystectomy for bladder cancer. They retrospectively reviewed the records of consecutive patients who underwent open radical cystectomy (ORC), laparoscopic radical cystectomy (LRC), or robot-assisted laparoscopic radical cystectomy (RARC) for bladder cancer at three medical centres in Japan. A total of 177 patients were enrolled. As a significant percentage of patients undergoing cystectomy are elderly, the authors investigated for an indicator that would determine the risk of surgery. Preoperative non-contrast computed tomography axial image at the third lumbar vertebra level was used to measure the mean Hounsfield unit and cross-sectional area (mm2 ) of the psoas muscle. Univariate analysis showed significant differences in age, sex, clinical T stage, and PMHU. In multivariate analysis using the Cox proportional hazards model, age (hazard ratio [HR]=1.734), sex (HR=2.116), cT stage (HR=1.665), and PMHU (HR=1.758) were significant predictors for overall survival. Furthermore, using these four predictors, it was possible to stratify the prognosis of patients after RC. Finally, PMHU was useful as a simple and significant preoperative factor that correlated with prognosis after the surgery.

The authors note that PMHU can be measured very simply. No special training or software is required. NCCT images are always acquired in patients before RC, without the burden of adding new special tests to the patients.

The authors believe that using the PMHU index could help to identify patients for whom radical cystectomy would carry too great a risk. In these patients, less invasive forms of treatment could be considered.

In my opinion, the publication is well planned and well written. It brings interesting information that may influence therapeutic decisions.

Author Response

Response to reviewers:

Thank the reviewers for these precious comments concerning my manuscript entitled Prognostic effect of preoperative psoas muscle Hounsfield unit at radical cystectomy for bladder cancer’. These comments are all valuable and extremely helpful for revising and improving our paper. We have checked comments carefully and have made corrections which we hope meet with approval. The responses to the reviewer’s comments are as follows:

Reviewer 2:

We should appreciate for your favorable comments. There are some corrections related to the answer to another reviewer, so we would appreciate it if you could check them.